# The Seasonal Variability of the Ocean Energy Cycle from a Quasi-Geostrophic Double Gyre Ensemble

**Takaya Uchida** *[ID], **Bruno Deremble** [ID] and **Thierry Penduff** [ID]

Université Grenoble Alpes, CNRS, IRD, Grenoble-INP, Institut des Géosciences de l'Environnement, CS 40 700, CEDEX 9, 38058 Grenoble, France; bruno.deremble@univ-grenoble-alpes.fr (B.D.); Thierry.PENDUFF@cnrs.fr (T.P.)
* Correspondence: takaya.uchida@univ-grenoble-alpes.fr

**Abstract:** With the advent of submesoscale $O(1\,\text{km})$ permitting basin-scale ocean simulations, the seasonality of mesoscale $O(50\,\text{km})$ eddies with kinetic energies peaking in summer has been commonly attributed to submesoscale eddies feeding back onto the mesoscale via an inverse energy cascade under the constraint of stratification and Earth's rotation. In contrast, by running a 101-member, seasonally forced, three-layer quasi-geostrophic (QG) ensemble configured to represent an idealized double-gyre system of the subtropical and subpolar basin, we find that the mesoscale kinetic energy shows a seasonality consistent with the summer peak without resolving the submesoscales; by definition, a QG model only resolves small Rossby and Froude number dynamics ($O(Ro) \ll 1, O(Fr) \ll 1$) while submesoscale dynamics are associated with $O(Ro) \sim 1, O(Fr) \gtrsim 1$. Here, by quantifying the Lorenz cycle of the mean and eddy energy, defined as the ensemble mean and fluctuations about the mean, respectively, we propose a different mechanism from the inverse energy cascade. During summer, when the Western Boundary Current is stabilized and strengthened due to increased stratification, stronger mesoscale eddies are shed from the separated jet. Conversely, the opposite occurs during the winter; the separated jet destablizes and results in overall lower mean and eddy kinetic energies despite the domain being more susceptible to baroclinic instability from weaker stratification.

**Keywords:** ocean circulation; Lorenz energy cycle; quasi-geostrophic flows; ensemble modelling

## 1. Introduction

The energy cycle of the atmospheric system, namely the energy exchange between the mean flow and fluctuations about the mean, have long been of interest due to the fluctuating flow being attributed to what is commonly known as the "weather" [1,2]. Similarly, the oceanographic community has had a long-standing interest in eddies, the weather system of the oceans [3–5]. In a seminal paper, Lorenz [2] provided a framework in understanding the eddy–mean flow interaction, a framework often referred to as the Lorenz energy cycle (herein LEC; [6]).

LEC generally decomposes the flow into four energy reservoirs: the mean and eddy available potential energy (APE) and kinetic energy (KE), respectively. The concept of APE is perhaps unique to the field of geophysical fluid dynamics where the gravitational force plays a dominant role in the governing equations. Although all geophysical fluids store gravitational potential energy, only a small fraction of it is available to generate fluid motion, hence the prefix "available". The energy exchanges between each reservoir elucidate the balance of physical processes responsible for causing the eddy flow [5], e.g., exchanges between the mean and eddy KE are associated with barotropic instability while exchanges between the eddy APE and eddy KE are associated with baroclinic instability. Barotropic instability is generated via horizontal shear in the mean flow while baroclinic instability occurs when the effect of gravity, due to weak vertical stratification, has a similar order of magnitude as the effects of Earth's rotation [7]. The balance between the two instabilities

results in the weather and eddies we commonly observe in the atmosphere and ocean. With the recent increase in computational power and advent of eddy resolving simulations of the ocean, there has been a growing interest in the interlinkage between the energy exchanges and temporal variability, namely seasonality, in the eddy flow [8,9].

In the context of Physical Oceanography, the eddies can be further separated into meso- and submesoscale eddies. Mesoscale eddies are roughly on the spatial scales of the first baroclinic Rossby radius of deformation ($NH/f \sim O(50\,\text{km})$ where $N$ and $H$ are the vertical stratification and ocean depth, respectively, and $f$ is the Coriolis parameter) while submesoscale eddies are on the scale of $O(1\,\text{km})$ [10]. In terms of the Rossby number ($Ro$ ($=U/f_0 L$)) and Froude number ($Fr$ ($=U/NH$)) where $U$ and $L$ are the characteristic scales of velocity and length, the spatial scales translate as mesoscale dynamics being on the order of $O(Ro) \ll 1, O(Fr) \ll 1$, and submesoscale flows being associated with $O(Ro) \sim 1, O(Fr) \gtrsim 1$ [11–16]. In other words, mesoscale dynamics are more constrained by Earth's rotation and stratification, leading to the well-known phenomenon of inverse energy cascade where KE is transferred from scales about the Rossby radius to larger scales [17–19]. To what extent the framework of inverse energy cascade is applicable for scales smaller than the Rossby radius remains an open question [11,12,20].

Although there is some geographical variability [21–25], many studies using meso- and submesoscale permitting ocean simulations have attributed the seasonality in mesoscale KE to energy being transferred upscale from the submesoscales where the seasonal modulation of the mixed-layer depth leads to a strong signal [14,26–31]. Instabilities within the mixed layer are inherently submesoscale due to the reduced stratification and shallow depth scale, and are most active during late winter/early spring when the mixed-layer is the deepest [32–34]. The summertime peak in mesoscale KE has consequently been explained by the time required for the submesoscale energy to cascade upscale. Other mechanisms, such as air–sea interaction, have also been argued for the cause of mesoscale seasonality [35]. While we agree that submesoscale instabilities and air–sea interaction affect mesoscale variability, here, we examine another mechanism on the other end of the spectrum in modulating the mesoscale seasonality: the basin-scale ($O(1000\,\text{km})$) affecting the mesoscale.

In order to quantify the exchanges between the energy reservoirs, we run a seasonally forced, three-layer quasi-geostrophic (QG) ensemble with a double-gyre configuration and examine the LEC. By definition, a QG model only resolves small Rossby number dynamics based on asymptotic expansion of the governing equations [36], i.e., the simulated eddy field only consists of mesoscale variability. The background state in quasi-geostrophy can be considered as the basin-scale state. In particular, we define the mean via the ensemble mean and eddies as the fluctuations about the mean. The ensemble mean: (i) negates the ergodic assumption where one treats the temporal mean equivalent to an ensemble mean, which is questionable for a temporally varying system; (ii) removes the arbitrary temporal and/or spatial scale in defining the mean [37]; (iii) is consistent with the Reynold's definition of eddy–mean decomposition [38]; and (iv) retains the temporal, namely seasonal, variability of the LEC.

The paper is organized as follows: We describe the model configuration in Section 2 and re-derive the layered QG equations and LEC in Section 3, which will aid our discussion later on. We present our results in Section 4 and conclude in Section 5.

## 2. Model Description

We use the quasi-geostrophic (QG) configuration of the Multiple Scale Ocean Model (MSOM; [39], herein referred to as MSQG), based on the Basilisk language [40], to simulate a three-layer double-gyre flow with a rigid lid and flat bottom. No-flux conditions are applied at the lateral boundaries. The parameters used are similar to prior QG studies which examine the dynamics of a double-gyre system [3,41,42] and are summarized in Table 1. The characteristic length scale of the Rossby radius (viz. radii of mesoscale eddies) is prescribed as $L$ ($=50\,\text{km}$) and horizontal resolution is $\sim 4\,\text{km}$ ($=\delta_{\hat{x}} L$) and therefore we have

roughly 12 grid points per radius; our simulation can be considered mesoscale resolving under the numerics of a second-order Arakawa advection scheme [43–45] (we note that our kilometric resolution does not allow for the submesoscales to be permitted in our model due to the QG constraint: $O(Ro) \ll 1, O(Fr) \ll 1$).

**Table 1.** Parameters used to configure the three-layer QG simulation and dimensionalized characteristic scales. The bottom Ekman number is the ratio between the bottom Ekman-layer thickness and $\widehat{H}_3$ and bottom friction is $\epsilon = Ek^b/(2Ro^m\widehat{H}_3)$. Beta is dimensionalized as $\beta = \hat{\beta}U/L^2$ and the dimensionalized domain size is 4000 km ($=\hat{L}_0 L$). The frequency of $Fr$ translates approximately to a 360-day year ($=\hat{f}_{Fr}^{-1}L/U$). The prognostic time stepping is determined via the CFL condition within values smaller than $\delta_{\hat{t}}^{\max}$.

| Parameter | Notation | Value | Unit |
|---|---|---|---|
| Number of horizontal grids | $N$ | 1024 | - |
| Number of vertical layers | $n_l$ | 3 | - |
| Non-dim. horizontal domain size | $\hat{L}_0$ | 80 | - |
| Non-dim. horizontal resolution | $\delta_{\hat{x}}$ | $N^{-1}\hat{L}_0$ | - |
| Background Rossby number | $Ro^m$ | 0.025 | - |
| Non-dim. Coriolis parameter | $\hat{f}_0$ | $Ro^{m-1}$ | - |
| Bottom Ekman number | $Ek^b$ | 0.004 | - |
| Non-dim. surface Ekman pumping | $\hat{\tau}_0$ | 0.0001 | - |
| Biharmonic Reynolds number | $Re_4$ | 4000 | - |
| Non-dim. beta | $\hat{\beta}$ | 0.5 | - |
| Background Froude number | $Fr_1^m; Fr_2^m$ | 0.00409959; 0.01319355 | - |
| Amplitude of $Fr_i$ | $\hat{A}_{Fr_1}; \hat{A}_{Fr_2}$ | 0.1; 0 | - |
| Non-dim. frequency of $Fr_i$ | $\hat{f}_{Fr_1}; \hat{f}_{Fr_2}$ | $62.2^{-1}; 62.2^{-1}$ | - |
| Non-dim. layer thickness | $\widehat{H}_1; \widehat{H}_2; \widehat{H}_3$ | 0.06; 0.14; 0.8 | - |
| Non-dim. reduced gravity | $\widehat{g}'_i$ | $Fr_i^{-2}\widehat{H_i^\dagger}$ | - |
| Non-dim. maximum time stepping | $\delta_{\hat{t}}^{\max}$ | $5 \times 10^{-2}$ | - |
| CFL condition | - | 0.4 | - |
| Horizontal velocity | $U$ | 0.1 | [m s$^{-1}$] |
| Length scale | $L$ | 50 | [km] |
| Total layer thickness | $H$ | 5000 | [m] |

MSQG solves prognostically for the non-dimensionalized QG potential vorticity (PV):

$$\frac{D\hat{q}}{D\hat{t}} = \widehat{\mathcal{F}} + \widehat{\mathcal{D}}, \tag{1}$$

where $q = \zeta_g + \beta y - \frac{f_0}{H}h$ is the QGPV (details are given in Appendix A; [7]) and the $\beta$-plane approximation is applied ($f = f_0 + \beta y$). $\mathcal{F}$ and $\mathcal{D}$ are the forcing and dissipative terms, and $\widehat{(\cdot)}$ are non-dimensionalized variables. The forcing term is the wind stress curl without any buoyancy forcing at the surface, and is kept stationary with the formulation:

$$\widehat{\mathcal{F}} = \frac{\widehat{\boldsymbol{\nabla}}_\mathrm{h} \times \hat{\boldsymbol{\tau}}(\hat{y})}{\widehat{H}_1} = -\frac{\hat{\tau}_0}{Ro^m\widehat{H}_1}\sin\left(\frac{2\pi}{N}\hat{y}\right)\sin\left(\frac{\pi}{N}\hat{y}\right), \tag{2}$$

where $\boldsymbol{\nabla}_\mathrm{h}$ is the horizontal gradient operator, and $\hat{y}$ ($\in [0.5, N - 0.5]$) is the non-dimensionalized meridional extent of the domain. Only the wind stress curl is prescribed in the model and not the wind stress itself ($\boldsymbol{\tau}$) but we denote it for clarity in notation. We have kept the wind stress curl axisymmetric as low-frequency variability is not the focus of this study [46–49]. The dissipation term is implemented via a biharmonic viscosity:

$$\widehat{\mathcal{D}} = -Re_4^{-1}\widehat{\boldsymbol{\nabla}}_\mathrm{h}^4\hat{q}. \tag{3}$$

The background stratification is defined at each layer interface via the Froude number where we enforce the seasonality by varying it in time according to:

$$Fr_i = \frac{U}{\sqrt{g_i' H_i^\dagger}} = Fr_i^m \left[1 + \hat{A}_{Fr_i} \sin\left(2\pi \hat{f}_{Fr_i} \hat{t}\right)\right]^{-1/2}, \tag{4}$$

where $H_i^\dagger = (H_i + H_{i+1})/2$, $g'$ is the reduced gravity and subscript $i$ is the layer index (Figure 1). We vary $Fr_1$ in time but keep $Fr_2$ stationary ($\hat{A}_{Fr_2} = 0$), which is consistent with the seasonal variability of stratification being confined in the upper few hundred meters in the real ocean [50].

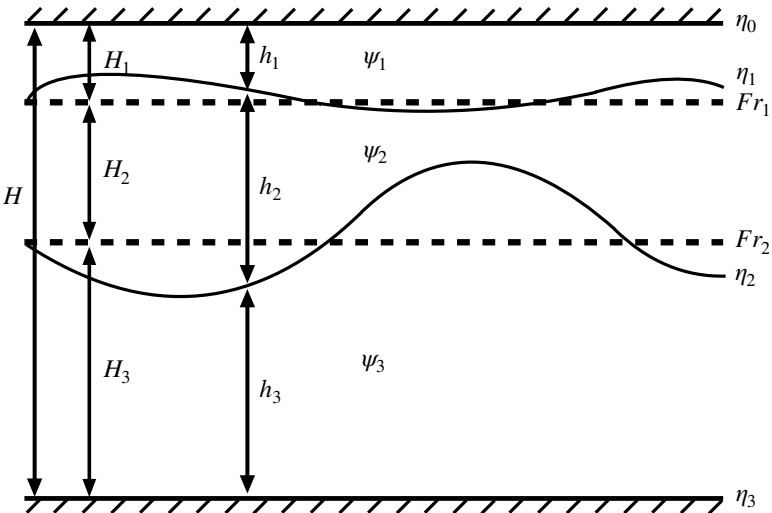

**Figure 1.** Vertical structure of the three-layer QG model with a rigid lid and flat bottom. The layer interface displacement ($\eta_i$) is shown in the thin curvy lines and net layer thickness is $h_i = H_i + \eta_{i-1} - \eta_i$. The stream functions ($\psi_i$) are defined within each layer.

We spin up the model for 10 years from a spun-up run with lower resolution ($N = 256$, equivalently $\delta_{\hat{x}} L \sim 15$ km) and then perturb the first-layer stream function at a single, random grid point with a perturbation on the order of ($O(10^{-5})$) to generate 100 slightly perturbed stream function fields. We use the perturbed fields as the initial conditions to generate 100 ensemble members. The surface wind stress and temporally varying background stratification are kept identical during the spin up and amongst ensemble members after the spin up. We run each ensemble member for another 10 years and for reference, we also have a control (CTRL) run without any perturbations to the initial condition; in total, we have 101 ensemble members and the CTRL run is there to show that the perturbations do not lead to a bifurcation in the dynamical regime within the 10 years of our simulation [51]. The model outputs were saved as instantaneous snapshots at every characteristic time scale ($\mathcal{T} = L/U = 5 \times 10^5$ seconds $\sim$5.8 days).

## 3. Derivation of the Lorenz Energy Cycle

Although the layered QG equations have been derived countless times [1,3,7,36], here, we re-derive the energy equations for a rigid-lid and flat-bottom three-layer QG model with a seasonally varying background stratification, in which the latter leads to some subtleties. In the remainder of the study, we only discuss dimensionalized variables. We start off with the order Rossby number relative vorticity equation for a given layer $i$ ($\in [1,3]$; Figure 1) neglecting viscous and external forcing terms:

$$\partial_t \zeta_{g;i} + u_{g;i} \partial_x \zeta_{g;i} + v_{g;i} \partial_y \zeta_{g;i} + \beta v_{g;i} = -f_0(\partial_x u_{a;i} + \partial_y v_{a;i})$$
$$= f_0 \partial_z w_{a;i}, \tag{5}$$

which are derived by taking the cross product of the momentum Equations (A6) and (A7). The subscripts $g$ and $a$ denote the geostrophic and ageostrophic components, respectively (e.g., $\zeta = \zeta_g + \zeta_a$). We denote the partial derivatives as $\partial_{(\cdot)}$ with respect to $(t, z, y, x)$. The stream function is defined as $\psi_i = \phi_{g;i}/f_0$ where $\phi_{g;i}$ is the geostrophic pressure anomaly and relative vorticity can be written as $\zeta_{g;i} = \nabla^2_{\mathrm{h}}\psi_i$. The layer-thickness equation on the other hand is [7]:

$$\partial_t h_i + u_{g;i}\partial_x h_i + v_{g;i}\partial_y h_i = -H_i(\partial_x u_{a;i} + \partial_y v_{a;i})$$
$$= H_i \partial_z w_{a;i} \tag{6}$$

We leave the derivation of the layered QGPV and its relation to the continuously stratified framework to Appendix A.

The ageostrophic vertical velocity can be diagnosed via the QG omega equation (Appendix B; [52,53]):

$$N_i^2 \nabla^2_{\mathrm{h}} w_{a;i} + f_0^2 \partial_{zz} w_{a;i} = \beta \partial_x b_i - 2\nabla_{\mathrm{h}} \cdot \mathbf{Q}_i - \nabla^2_{\mathrm{h}} b_i N_i^2 \partial_t \frac{1}{N_i^2}, \tag{7}$$

where $N_i^2 = g_i'/H_i^{\dagger}$, and $b_i = f_0 \frac{\psi_i - \psi_{i+1}}{H_i^{\dagger}}$ is the buoyancy. The $\mathbf{Q}$ tensor is:

$$\mathbf{Q}_i = Q_i^1 \mathbf{i} + Q_i^2 \mathbf{j} = \left(\partial_x \mathbf{u}_{g;i}^{\dagger} \cdot \nabla_{\mathrm{h}} b_i\right)\mathbf{i} + \left(\partial_y \mathbf{u}_{g;i}^{\dagger} \cdot \nabla_{\mathrm{h}} b_i\right)\mathbf{j}, \tag{8}$$

where $\mathbf{u}_{g;i}^{\dagger} = -\partial_y \psi_i^{\dagger} \mathbf{i} + \partial_x \psi_i^{\dagger} \mathbf{j}$ is the geostrophic velocity derived from the inter-facial stream function ($\psi_i^{\dagger} = \frac{H_i \psi_{i+1} + H_{i+1}\psi_i}{H_i + H_{i+1}}$; [3]). $\mathbf{i}$ and $\mathbf{j}$ are the horizontal Cartesian unit vectors. The last term on the right-hand side of (7) is due to the temporally varying background stratification (Appendix B). We solved Equation (7) iteratively for $w_a$ via a two-dimensional geometric multigrid solver with the boundary conditions of Ekman pumping ($w_E$):

$$w_{E;0} = -\frac{1}{f_0}\nabla_{\mathrm{h}} \times \boldsymbol{\tau} = -\frac{UH}{L}\hat{\tau}_0 \sin^2\left[\frac{2\pi y}{L_0}\right]\sin\left[\frac{\pi y}{L_0}\right], \tag{9}$$

$$w_{E;3} = \frac{\delta_E}{2}\zeta_{g;3}, \tag{10}$$

where $\delta_E = Ek^b H_3$ is the bottom Ekman-layer thickness.

Now, multiplying Equation (5) by $-\psi_i$ and integrating over the depth of each layer gives the kinetic energy (KE) budget:

$$H_i\left[\frac{D_i}{Dt}\frac{|\nabla_{\mathrm{h}}\psi_i|^2}{2} - \nabla_{\mathrm{h}} \cdot (\mathbf{u}_{g;i}\psi_i \nabla^2_{\mathrm{h}}\psi_i) - \frac{\beta}{2}\partial_x \psi_i^2\right]$$
$$= -f_0 \int \psi_i \partial_z w_{a;i}\, dz$$
$$= f_0\left[-(w_{a;i-1}\psi_{i-1}^{\dagger} - w_{a;i}\psi_i^{\dagger}) + \int w_{a;i}\partial_z \psi_i\, dz\right]$$
$$= f_0\left[-(w_{a;i-1}\psi_{i-1}^{\dagger} - w_{a;i}\psi_i^{\dagger}) + H_i\left(w_{a;i-1}\frac{\psi_{i-1} - \psi_i}{H_i + H_{i-1}} + w_{a;i}\frac{\psi_i - \psi_{i+1}}{H_{i+1} + H_i}\right)\right]. \tag{11}$$

Dropping the divergence terms as they vanish upon area integration, for each layer, we get:

$$\frac{H_1}{2}\partial_t |\nabla_{\mathrm{h}}\psi_1|^2 = f_0\left[w_{a;1}\psi_1^{\dagger} + w_{a;1}H_1\frac{\psi_1 - \psi_2}{H_2 + H_1}\right], \tag{12}$$

$$\frac{H_2}{2}\partial_t |\nabla_{\mathrm{h}}\psi_2|^2 = f_0\left[-(w_{a;1}\psi_1^{\dagger} - w_{a;2}\psi_2^{\dagger}) + H_2\left(w_{a;1}\frac{\psi_1 - \psi_2}{H_2 + H_1} + w_{a;2}\frac{\psi_2 - \psi_3}{H_3 + H_2}\right)\right], \tag{13}$$

$$\frac{H_3}{2}\partial_t|\boldsymbol{\nabla}_h\psi_3|^2 = f_0\Big[-w_{a;2}\psi_2^\dagger + w_{a;2}H_3\frac{\psi_2-\psi_3}{H_3+H_2}\Big]. \tag{14}$$

On the other hand, using relation (A2), the layer-thickness equations can be manipulated as $\frac{H_2}{H_1+H_2}(6)|_{i=1} - \frac{H_1}{H_1+H_2}(6)|_{i=2}$:

$$\begin{aligned}
\frac{D_1^\dagger}{Dt}\Big[\frac{f_0}{g_1'}(\psi_1-\psi_2)\Big] &= -w_{a;1} + \frac{H_1}{H_1+H_2}\Big[w_{a;2} - \frac{D_2}{Dt}\frac{f_0}{g_2'}(\psi_3-\psi_2)\Big] \\
&= -w_{a;1} + \frac{f_0 H_1}{g_2'(H_1+H_2)}(\boldsymbol{u}_3-\boldsymbol{u}_2)\cdot\boldsymbol{\nabla}_h(\psi_3-\psi_2) \\
&= -w_{a;1},
\end{aligned} \tag{15}$$

where the second term on the right-hand side above (15) vanishes due to thermal wind. Similarly, $\frac{H_3}{H_2+H_3}(6)|_{i=2} - \frac{H_2}{H_2+H_3}(6)|_{i=3}$:

$$\begin{aligned}
\frac{D_2^\dagger}{Dt}\Big[\frac{f_0}{g_2'}(\psi_2-\psi_3)\Big] &= -w_{a;2} + \frac{H_3}{H_2+H_3}\Big[w_{a;1} - \frac{D_2}{Dt}\frac{f_0}{g_1'}(\psi_2-\psi_1)\Big] \\
&= -w_{a;2},
\end{aligned} \tag{16}$$

where $\frac{D_i^\dagger}{Dt} = \partial_t + \boldsymbol{u}_{g;i}^\dagger\cdot\boldsymbol{\nabla}_h$. The available potential energy (APE) equations can, therefore, be derived by multiplying Equation (15) with $f_0(\psi_1-\psi_2)$ and again dropping the divergence terms:

$$\partial_t\Big[\frac{f_0^2}{2g_1'}(\psi_1-\psi_2)^2\Big] = -f_0(\psi_1-\psi_2)w_{a;1} - \frac{f_0^2(\psi_1-\psi_2)^2}{2}\partial_t g_1'^{-1}, \tag{17}$$

and Equation (16) with $f_0(\psi_2-\psi_3)$:

$$\partial_t\Big[\frac{f_0^2}{2g_2'}(\psi_2-\psi_3)^2\Big] = -f_0(\psi_2-\psi_3)w_{a;2}. \tag{18}$$

We see from Equation (17) that there is an additional source of APE due to the temporally varying background potential energy (BPE; $B^\#$), which then feeds back onto the KE via Equations (12) and (13) through baroclinic instability. BPE takes the same form as APE except that only $g'$ is inside the derivative.

Now, the mean KE (MKE; $K^\#$), eddy KE (EKE; $\mathcal{K}$), mean APE (MAPE; $P^\#$) and eddy APE (EAPE; $\mathcal{P}$) can be defined as:

$$K_i^\# = \frac{H_i}{2}|\boldsymbol{\nabla}_h\overline{\psi_i}|^2, \quad \mathcal{K}_i = \frac{H_i}{2}\overline{|\boldsymbol{\nabla}_h\psi_i'|^2}, \tag{19}$$

$$P_i^\# = \frac{f_0^2}{2g_i'}(\overline{\psi_i}-\overline{\psi_{i+1}})^2, \quad \mathcal{P}_i = \frac{f_0^2}{2g_i'}\overline{(\psi_i'-\psi_{i+1}')^2}, \tag{20}$$

where $\overline{(\cdot)}$ is the ensemble mean and the eddy is defined as fluctuations about the ensemble mean, viz. $(\cdot)' = (\cdot) - \overline{(\cdot)}$. We note that the ensemble mean of the fluctuations vanish $(\overline{(\cdot)'} = 0)$. The strength of defining the mean as such is that in addition to the ensemble-mean operator commuting with the derivatives with respect to $(t, z, y, x)$ [38], it provides a unique decomposition between the mean and eddy. In other words, the mean does not depend on an arbitrary temporal or spatial scale, which is beneficial in our case as the separated jet is on QG scaling in the cross-jet direction while on planetary-geostrophic scaling in the along-jet direction [54,55]. The ensemble mean can be interpreted as the QG response to external forcing while the eddies are a result of intrinsic variability [56–58]. The ensemble means of total KE and APE each satisfy $\overline{K_i} = \frac{H_i}{2}\overline{|\nabla_h\psi_i|^2} = K_i^\# + \mathcal{K}_i$, $\overline{P_i} = $

$\frac{f_0^2}{2g_i'}\overline{(\psi_i - \psi_{i+1})^2} = P_i^\# + \mathcal{P}_i$. Hence, the exchanges ($\Pi$) of KE and APE within and between layers are:

$$\Pi_{K_1^\# \to \mathcal{K}_1} = -H_1 \langle \overline{\psi_1} \boldsymbol{\nabla}_{\mathrm{h}} \cdot \overline{\boldsymbol{u}_{g;1}' \boldsymbol{\nabla}_{\mathrm{h}}^2 \psi_1'} \rangle, \tag{21}$$

$$\Pi_{K_1^\# \to K_2^\#} = -f_0 \langle \overline{w_{a;1}} \overline{\psi_1^\dagger} \rangle, \quad \Pi_{\mathcal{K}_1 \to \mathcal{K}_2} = -f_0 \langle \overline{w_{a;1}' \psi_1^{\dagger\prime}} \rangle, \tag{22}$$

$$\Pi_{P_1^\# \to K_1^\#} = \frac{f_0 H_1}{H_2 + H_1} \langle \overline{w_{a;1}}(\overline{\psi_1} - \overline{\psi_2}) \rangle, \quad \Pi_{\mathcal{P}_1 \to \mathcal{K}_1} = \frac{f_0 H_1}{H_2 + H_1} \langle \overline{w_{a;1}'(\psi_1' - \psi_2')} \rangle, \tag{23}$$

$$\Pi_{P_1^\# \to \mathcal{P}_1} = \frac{f_0^2}{g_1'} \langle (\overline{\psi_1} - \overline{\psi_2}) \boldsymbol{\nabla}_{\mathrm{h}} \cdot \overline{\boldsymbol{u}_{g;1}^{\dagger\prime}(\psi_1' - \psi_2')} \rangle, \tag{24}$$

$$\Pi_{B_1^\# \to P_1^\#} = -\frac{f_0^2}{2} \langle (\overline{\psi_1} - \overline{\psi_2})^2 \partial_t {g_1'}^{-1} \rangle, \quad \Pi_{B_1^\# \to \mathcal{P}_1} = -\frac{f_0^2}{2} \langle \overline{(\psi_1' - \psi_2')^2} \partial_t {g_1'}^{-1} \rangle, \tag{25}$$

$$\Pi_{K_2^\# \to \mathcal{K}_2} = -H_2 \langle \overline{\psi_2} \boldsymbol{\nabla}_{\mathrm{h}} \cdot \overline{\boldsymbol{u}_{g;2}' \boldsymbol{\nabla}_{\mathrm{h}}^2 \psi_2'} \rangle, \tag{26}$$

$$\Pi_{K_2^\# \to K_3^\#} = -f_0 \langle \overline{w_{a;2}} \overline{\psi_2^\dagger} \rangle, \quad \Pi_{\mathcal{K}_2 \to \mathcal{K}_3} = -f_0 \langle \overline{w_{a;2}' \psi_2^{\dagger\prime}} \rangle, \tag{27}$$

$$\Pi_{P_1^\# \to K_2^\#} = \frac{f_0 H_2}{H_2 + H_1} \langle \overline{w_{a;1}}(\overline{\psi_1} - \overline{\psi_2}) \rangle, \quad \Pi_{\mathcal{P}_1 \to \mathcal{K}_2} = \frac{f_0 H_2}{H_2 + H_1} \langle \overline{w_{a;1}'(\psi_1' - \psi_2')} \rangle, \tag{28}$$

$$\Pi_{P_2^\# \to K_2^\#} = \frac{f_0 H_2}{H_3 + H_2} \langle \overline{w_{a;2}}(\overline{\psi_2} - \overline{\psi_3}) \rangle, \quad \Pi_{\mathcal{P}_2 \to \mathcal{K}_2} = \frac{f_0 H_2}{H_3 + H_2} \langle \overline{w_{a;2}'(\psi_2' - \psi_3')} \rangle, \tag{29}$$

$$\Pi_{P_2^\# \to \mathcal{P}_2} = \frac{f_0^2}{g_2'} \langle (\overline{\psi_2} - \overline{\psi_3}) \boldsymbol{\nabla}_{\mathrm{h}} \cdot \overline{\boldsymbol{u}_{g;2}^{\dagger\prime}(\psi_2' - \psi_3')} \rangle, \tag{30}$$

$$\Pi_{K_3^\# \to \mathcal{K}_3} = -H_3 \langle \overline{\psi_3} \boldsymbol{\nabla}_{\mathrm{h}} \cdot \overline{\boldsymbol{u}_{g;3}' \boldsymbol{\nabla}_{\mathrm{h}}^2 \psi_3'} \rangle, \tag{31}$$

$$\Pi_{P_2^\# \to K_3^\#} = \frac{f_0 H_3}{H_2 + H_3} \langle \overline{w_{a;2}}(\overline{\psi_2} - \overline{\psi_3}) \rangle, \quad \Pi_{\mathcal{P}_2 \to \mathcal{K}_3} = \frac{f_0 H_3}{H_3 + H_2} \langle \overline{w_{a;2}'(\psi_2' - \psi_3')} \rangle, \tag{32}$$

where $\langle \cdot \rangle = \iint (\cdot) dx dy$ is the area integration. Further details regarding the sign convention and forcing/dissipation terms are given in Appendices C and D. Summing up each layer gives the volume integrated energy exchanges:

$$\Pi_{P^\# \to K^\#} = \sum_{i=1}^{2} f_0 \langle \overline{w_{a;i}}(\overline{\psi_i} - \overline{\psi_{i+1}}) \rangle, \tag{33}$$

$$\Pi_{\mathcal{P} \to \mathcal{K}} = \sum_{i=1}^{2} f_0 \langle \overline{w_{a;i}'(\psi_i' - \psi_{i+1}')} \rangle, \tag{34}$$

$$\Pi_{P^\# \to \mathcal{P}} = \sum_{i=1}^{2} \frac{f_0^2}{g_i'} \langle (\overline{\psi_i} - \overline{\psi_{i+1}}) \boldsymbol{\nabla}_{\mathrm{h}} \cdot \overline{\boldsymbol{u}_{g;i}^{\dagger\prime}(\psi_i' - \psi_{i+1}')} \rangle, \tag{35}$$

$$\Pi_{K^\# \to \mathcal{K}} = -\sum_{i=1}^{3} H_i \langle \overline{\psi_i} \boldsymbol{\nabla}_{\mathrm{h}} \cdot \overline{\boldsymbol{u}_{g;i}' \boldsymbol{\nabla}_{\mathrm{h}}^2 \psi_i'} \rangle, \tag{36}$$

$$\Pi_{B^\# \to P^\#} = -\frac{f_0^2}{2} \langle (\overline{\psi_1} - \overline{\psi_2})^2 \partial_t {g_1'}^{-1} \rangle, \tag{37}$$

$$\Pi_{B^\# \to \mathcal{P}} = -\frac{f_0^2}{2} \langle \overline{(\psi_1' - \psi_2')^2} \partial_t {g_1'}^{-1} \rangle. \tag{38}$$

## 4. Results

We start by showing the total kinetic energy (TKE) during the spin-up phase and for the 10 years of output we have (viz. 20 years in total; Figure 2). The ensemble spread starts to grow after 1.5 years of integration from the perturbed initial conditions and plateaus

roughly for the latter seven years. The area-integrated TKE in the first layer ($\langle K_1 \rangle$), most relevant for studies interested in surface seasonal dynamics, is in sync with the background stratification ($g_1'$), viz. higher $\langle K_1 \rangle$ during summer when stratification is stronger and visa versa (Figure 2b). For the lower layers, there is a temporal lag evident by the barotropic TKE ($\langle |\boldsymbol{\nabla}_\mathrm{h} \Psi|^2 \rangle$ where $\Psi = H^{-1} \sum_i H_i \psi_i$ is the barotropic stream function; Figure 2a). Although it is difficult to detect a clear seasonal signal for the barotropic TKE from an individual ensemble member such as in the CTRL run, their ensemble mean shows a robust seasonality. For the remainder of the study, we use the last five years of output in order to maximize the signal of intrinsic variability amongst members.

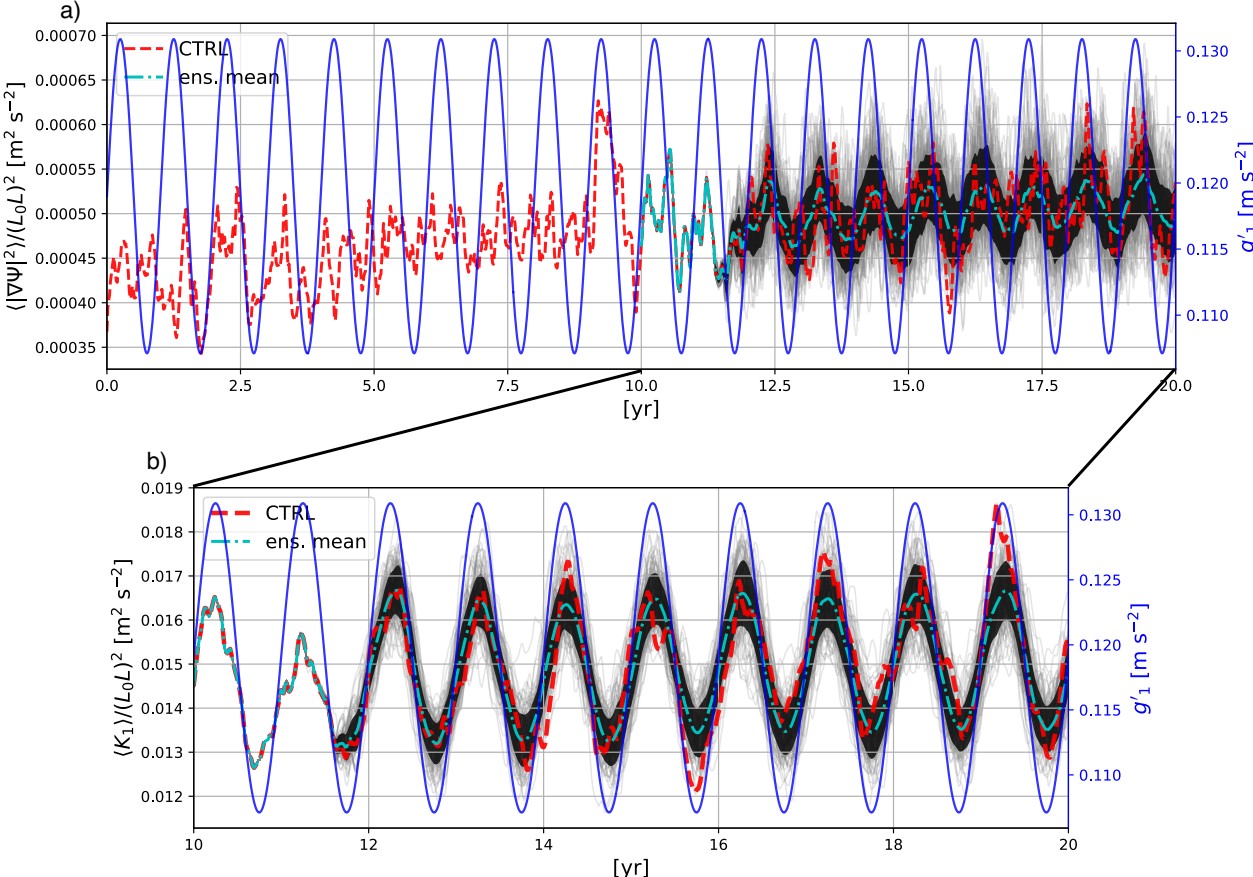

**Figure 2.** Time series of the horizontally averaged barotropic (**a**) and first-layer TKE (**b**). Each ensemble member is shown with thin gray lines and standard deviation of the ensemble mean in black shading. The CTRL run is shown with a red dashed line and ensemble mean with a cyan dot-dashed line. The reduced gravity is shown in blue plotted against the right *y* axis.

In Figure 3, we show the mean and eddy KE in the first layer ($K_1^\#, \mathcal{K}_1$) during summer and winter for the last year of output and their difference. The seasons were defined at the time step when the reduced gravity was at its maximum and minimum, respectively. We see the characteristic feature of a robust separated Western Boundary Current in a double-gyre system with very little meandering while the EKE is more meridionally spread out. Consistent with Figure 2, summertime has a stronger mean jet and EKE than winter (Figure 3e,f). We also show snapshots of eddy PV ($q_{g;1}'$) from the CTRL run from which we see coherent features of mesoscale eddies (Figure 3g,h).

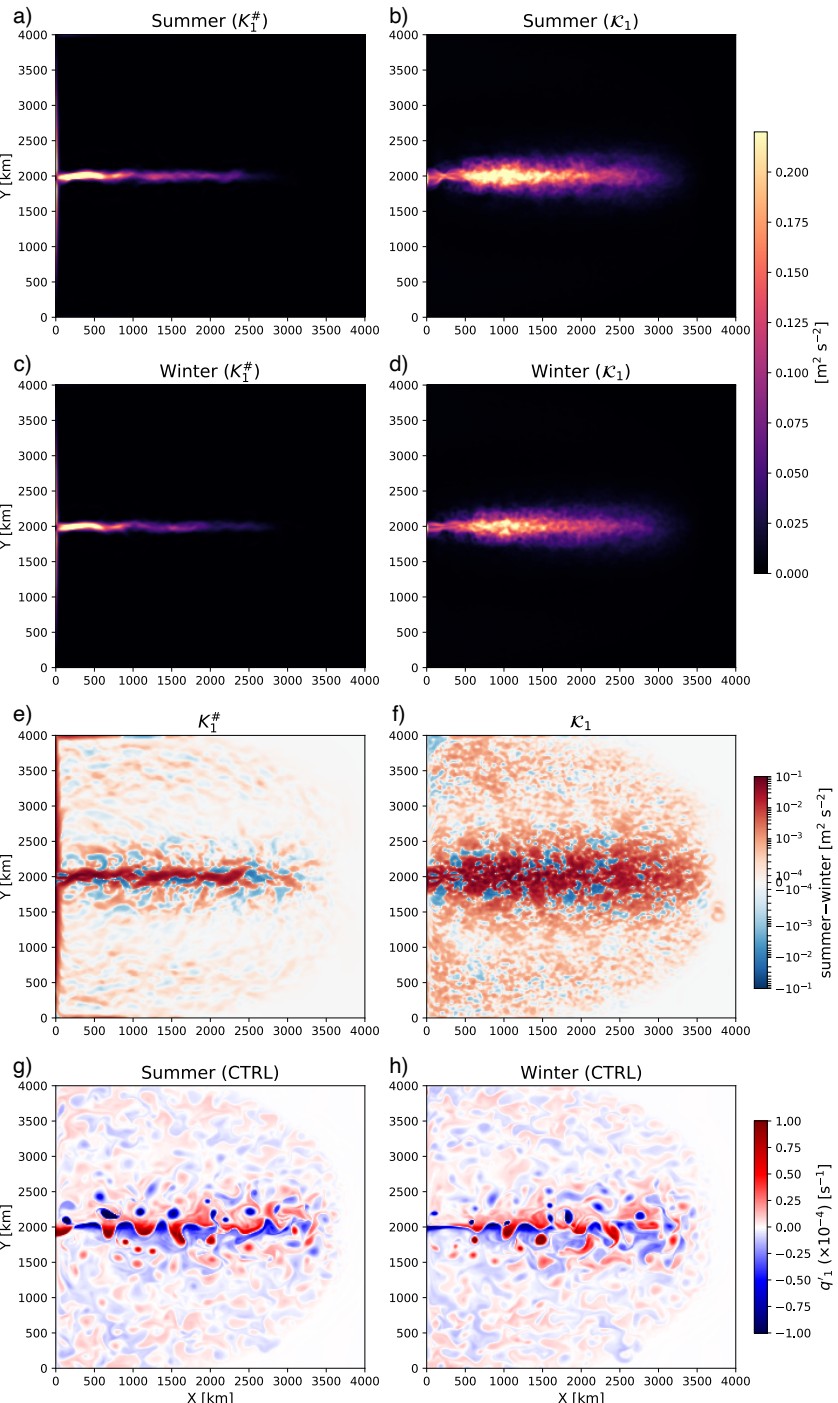

**Figure 3.** The summer and wintertime mean and eddy KE and their difference during the last year of output (**a–f**). Note the differences are plotted on a logarithmic scale. (**g,h**) Snapshot of eddy PV for summer and winter during the last year of output from the CTRL run. All panels show the variable in the first layer.

*4.1. The Domain Integrated Lorenz Energy Cycle*

We now move on to quantifying the LEC in order to examine the processes responsible for generating higher KE during summertime. As we define the mean as the ensemble mean (as opposed to a temporal mean which has commonly been applied), we are able to examine the temporal variability of LEC. We compute the terms in Equations (33)–(38) for the last five years of output and show them in Figure 4. The time series of MAPE is in sync with the background stratification dominated by $g'$ in its denominator while MKE

lags $g_1'$ by $\sim$11 days upon taking the lag correlation ($P^\#$, $K^\#$; Figure 4a). MAPE has the largest magnitude amongst the reservoirs by an order of magnitude and for KE, the eddies are more energetic than the mean. The energy flux from MAPE to MKE is negative year round ($\Pi_{P^\#\to K^\#} < 0$; black solid line in Figure 4b) due to Ekman pumping steepening the isopycnals. The energy input due to wind stress ($F_s^{K^\#}$) is in sync with MKE with energetic surface currents resulting in a stronger surface stress. The eddy energy reservoirs ($\mathcal{P}$, $\mathcal{K}$), on the other hand, lag the stratification by $\sim$17 days but their peaks precede winter when the domain is most susceptible to baroclinic instability and energy conversion from EAPE to EKE takes its yearly maximum ($\Pi_{\mathcal{P}\to\mathcal{K}}$; Figure 4a). It is perhaps interesting to note that the sign of flux between EAPE and EKE occasionally reverses during summer with barotropic instability over compensating for baroclinic instability; the energy pathway becomes MKE$\to$EKE$\to$EAPE ($\Pi_{\mathcal{P}\to\mathcal{K}} < 0$), whereas baroclinic instability would predict EAPE$\to$EKE ($\Pi_{\mathcal{P}\to\mathcal{K}} > 0$). Regarding the dissipation terms, only the bottom drag for EKE ($D_b^{\mathcal{K}}$) shows a notable seasonality and has a similar magnitude to the energy flux from MKE to EKE ($\Pi_{K^\#\to\mathcal{K}}$). The amplitude of bottom drag ($|D_b^{\mathcal{K}}|$) lags EKE by $\sim$41 days and aligns well with the ensemble-mean barotropic TKE (Figure 2a).

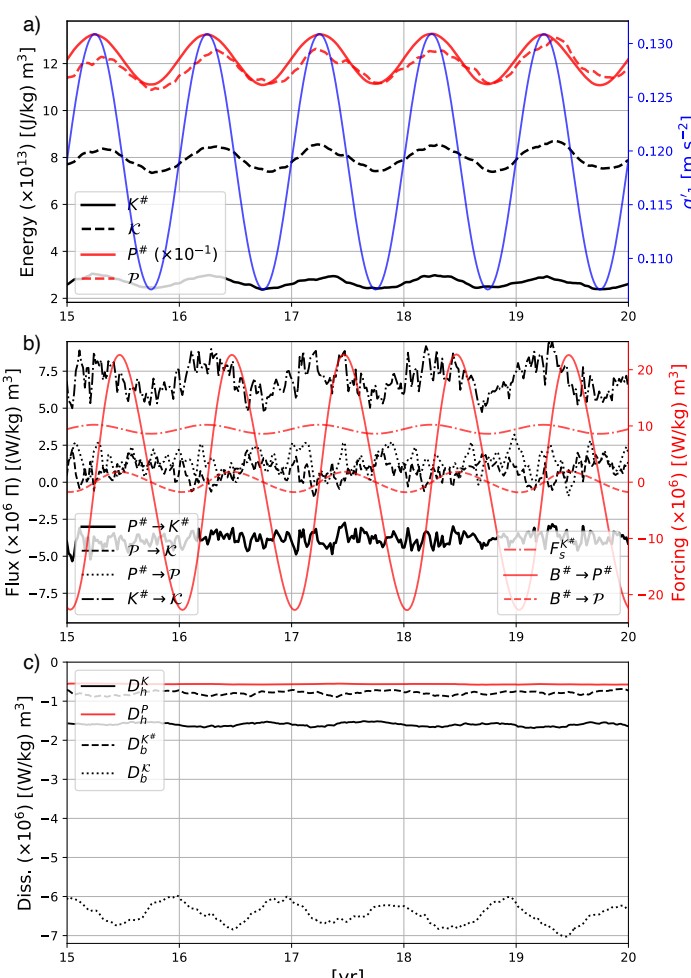

**Figure 4.** Time series of each term in the domain-integrated LEC. (**a**) The mean and eddy KE (black) and APE (red) reservoirs in the units of $\times 10^{13}$ [(J/kg) m$^3$] and stratification of the first layer interface (blue; $g_1'$). MAPE is multiplied by 0.1 to have it fit on the same $y$ axis. (**b**) The energy fluxes between each energy reservoir and forcing terms due to surface wind stress ($F_s^{K^\#}$) and temporally varying BPE. (**c**) Dissipation terms due to horizontal viscosity ($D_h$) and bottom friction ($D_b$). The mean and eddy horizontal dissipation terms are lumped together and fluxes are in the units of $\times 10^6$ [(W/kg) m$^3$]. The forcing and dissipation terms are detailed in Appendix D.

To provide a climatological view of the energy fluxes ($\Pi$), we take the yearly average of the last five years and show the LEC diagram for a climatological summer and winter (Figure 5). Each season per year is defined as four time steps; summer is when the reduced gravity takes its maxima and four time steps about it, and four time steps about the minima in reduced gravity for winter. The seasonal climatology is then taken as the average of the five years. Again, we see that all reservoirs are more energetic during the summer. Focusing on MKE, except for the surface wind stress, the reservoir has loss terms year round and yet stores more energy during the summer. We attribute the summertime maxima in MKE to the separated jet stabilizing due to increased stratification, which results in the jet shedding stronger eddies. Indeed, the energy flux from MKE to EKE ($\Pi_{K^\# \to \mathcal{K}}$) is highest during the summer (Figures 4b and 5a). We attribute the larger energy conversion from EAPE to EKE during the winter ($\Pi_{\mathcal{P} \to \mathcal{K}}$) to the flow being more susceptible to baroclinic instability with reduced stratification.

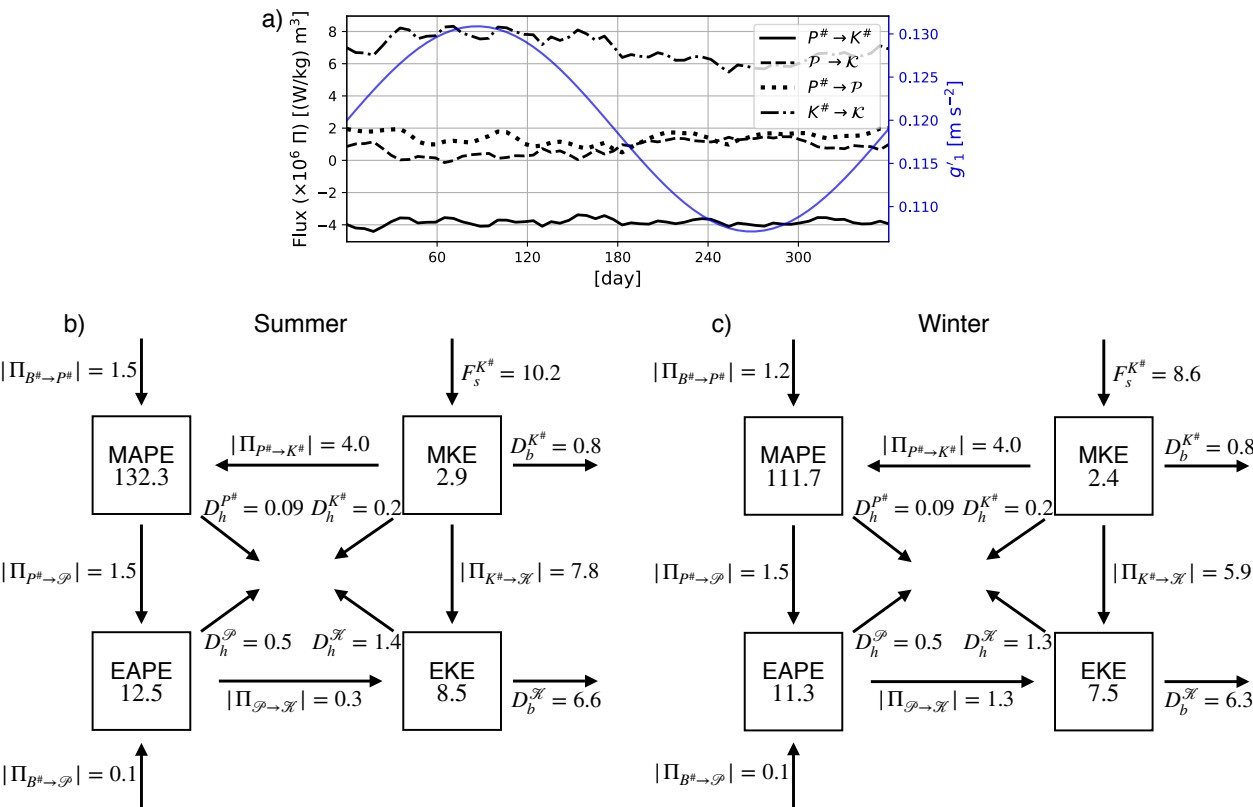

**Figure 5.** Time series of the seasonal climatology of energy fluxes between the energy reservoirs (**a**). (**b**,**c**) The LEC diagram for the climatological summer and winter averaged over the last five years of output. The energies are in the units of $\times 10^{13}$ [(J/kg) m$^3$] and fluxes are in $\times 10^6$ [(W/kg) m$^3$]. The energy exchanges do not exactly cancel out due to each reservoir having temporal variability.

### 4.2. Time Lag in Lower-Layer Energetics

In this section, we investigate the mechanism for the lag in KE in the lower layers ($K_2$, $K_3$) from KE in the first layer ($K_1$) and stratification ($g'_1$) implied from Figure 2. It is perhaps interesting to note that although the ensemble-mean barotropic TKE lags $g'_1$ by $\sim$41 days, neither the domain-integrated MKE nor EKE show such a long lag (Figure 4a). This has to do with MKE and EKE being volume-weighted variables of quadratic terms, while the barotropic TKE being a quadratic term of a volume-weighted variable; MKE and EKE have a larger weighting on the surface stream function, which is in sync with $g'_1$, than the barotropic TKE. The lag within lower layers becomes apparent for the time series of area integrated EKE within each layer ($\sim$128 days for $\mathcal{K}_2$ and $\sim$68 days for $\mathcal{K}_3$; Figure 6a). We also focus on EKE for the remainder of this section as EKE is always larger than MKE

by a factor of three (Figure 4). Examining the energy fluxes, Figure 6b shows that the contribution from barotropic instability becomes negligible within the lower two layers with the relative significance of the separated jet diminishing with depth, and shows no clear seasonality ($\Pi_{K^{\#}_{2,3} \to \mathcal{K}_{2,3}}$). The vertical transfer of EKE ($\Pi_{\mathcal{K}_1 \to \mathcal{K}_2}$, $\Pi_{\mathcal{K}_2 \to \mathcal{K}_3}$) and conversion from EAPE ($\Pi_{\mathcal{P} \to \mathcal{K}}$), on the other hand, show a coherent seasonal pattern with the maxima of $\mathcal{K}_2$ and $\mathcal{K}_3$ falling in between the maxima of the two fluxes. We, therefore, attribute the time lag in $\mathcal{K}_2$ and $\mathcal{K}_3$ to the balance between baroclinic instability and vertical transfers of EKE.

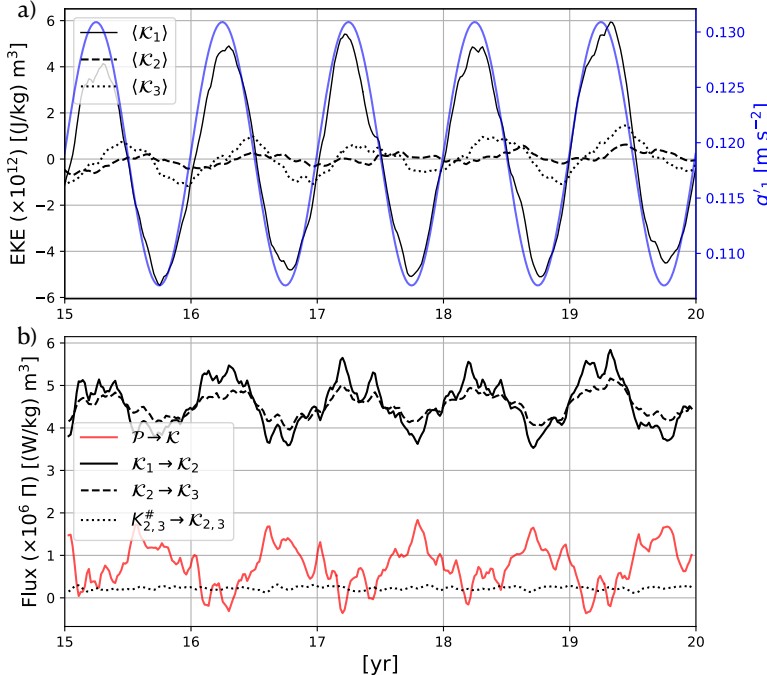

**Figure 6.** Time series of volume-integrated EKE over each layer, and fluxes within and between layers ($\Pi_{\mathcal{K}_1 \to \mathcal{K}_2}$, $\Pi_{\mathcal{K}_2 \to \mathcal{K}_3}$) plotted along with the reduced gravity ($g'_1$). (**a**) The EKEs have their temporal mean removed so as to plot against the same *y* axis. (**b**) A rolling mean by five time steps (∼29 days) is applied to the time series of the energy fluxes. The energy flux from MKE to EKE within the two bottom layers is summed up ($\Pi_{K^{\#}_{2,3} \to \mathcal{K}_{2,3}}$) and conversion from APE is shown as the conversion rate volume integrated over the three layers as the amount that goes into each layer is simply the total conversion weighted by layer thickness (cf. Equations (23), (28), (29) and (32)). The conversion from $\mathcal{P}_1$ and $\mathcal{P}_2$ were in sync with each other (not shown). For further details regarding each term, see Appendix D.

## 5. Discussion and Conclusions

By running a seasonally forced 101-member ensemble of a three-layer quasi-geostrophic (QG) model in an idealized double-gyre configuration, we have shown that the kinetic energy (KE) peaks during summer when the (basin-scale) stratification is strongest during the year (Figure 2). Such seasonality in mesoscale eddy KE (EKE) has been observed in other studies using realistic simulations of the ocean [14,27–31]. Due to the air–sea interaction, the seasonal modulation of the mixed-layer depth leads to a strong seasonal signal in submesoscale instabilities. The submesoscale EKE takes its maximum during late winter/early spring and previous studies have commonly explained the summer peak in the mesoscale range as the time lag for the submesoscale EKE to cascade upscale. The mechanism of inverse energy cascade fails, however, to explain the mesoscale seasonality in our model, as a QG model by definition cannot resolve any submesoscale instabilities.

Using the framework of the Lorenz energy cycle (LEC; [2]), we have quantified the reservoirs of mean and eddy available potential energy (APE) and KE, and energy fluxes amongst them. We note that our ensemble framework has allowed us to examine the

seasonal variability of LEC. Our results show that all four reservoirs store more energy during the summer than winter (Figure 4a). For the mean KE (MKE), we attribute the summertime maximum to increased stratification leading to a more baroclinically stable and stronger jet. Conceptually, this can be understood based on a mass–flux balance. Since the wind stress is kept stationary, the Sverdrup transport ($\beta^{-1}\boldsymbol{\nabla}_{\mathrm{h}} \times \boldsymbol{\tau}$) remains constant throughout the simulation. Based on mass balance, the accumulating transport towards the north/south boundaries must be fluxed out via the Western Boundary Current. Figure 3c shows an intensification of MKE during summer along the Western Boundary resulting from less energy lost to the eddies within the gyre interior. Hence, a more stable jet results in a stronger mean flow. Our results of jet stabilization and its zonal penetration into the gyre are complementary to earlier studies where they attributed the penetration scale to parameters of lateral friction, vertical resolution and topography [4,59]. Here, we have investigated the effect of a seasonally varying background stratification.

Shifting our focus to EKE, based on baroclinic instability, one might expect the opposite to be true, namely, wintertime having more EKE than summertime due to weaker stratification. The LEC shows that year round, energy fluxes from MKE to EKE associated with barotropic instability over compensate for the fluxes from eddy APE (EAPE) to EKE, a pathway associated with baroclinic instability. Since MKE is higher during summer, the larger flux of energy from MKE to EKE results in EKE peaking in summer (Figures 4b and 5). Although our simulation is highly idealized, we argue that barotropic processes dominating in the separated jet region are consistent with a recent study on energetics using a realistic simulation of the North Atlantic Ocean [55]. We note that the balance between barotropic and baroclinic instability in our LEC is in the domain integrated sense. In a domain without a jet, we would expect baroclinic instability to be the dominant mechanism in generating eddies so long as the background state is baroclinically unstable.

To our knowledge, Qiu et al. [26] is the only study using a realistic ocean simulation showing how the seasonality in background state can modulate the mesoscale variability. Their results differ from ours, however, in that they attribute the mesoscale seasonality to the classical Phillips-like baroclinic instability arising from the interior background stratification and vertical shear in horizontal velocity [1]. In addition to the submesoscale variability modulating mesoscale seasonality, our results suggest that, in reality, it is possible that the basin-scale variability does so as well. We note that since our QG model does not permit submesoscales, the baroclinic energy flux from EAPE to EKE is likely underestimated compared to the real ocean. It would be interesting to revisit the LEC for realistic ocean ensembles [57,58] to see whether we would see a stabilization of the separated Gulf Stream during summer and consequently larger energy fluxes from MKE to EKE.

**Author Contributions:** Conceptualization, T.U.; methodology, T.U. and B.D.; software, B.D.; validation, T.U., B.D.; formal analysis, T.U.; investigation, T.U. and B.D.; computational resources, T.P.; data curation, T.U.; writing, T.U.; visualization, T.U.; project administration, T.P.; funding acquisition, T.P. All authors have read and agreed to the published version of the manuscript.

**Funding:** This research was funded by the French 'Make Our Planet Great Again' (MOPGA) initiative managed by the Agence Nationale de la Recherche under the Programme d'Investissement d'Avenir, with the reference ANR-18-MPGA-0002. We acknowledge high-performance computing resources for generating the ensemble and analyzing our model outputs on Occigen maintained by CINES with the reference A0090112020.

**Institutional Review Board Statement:** Not applicable.

**Informed Consent Statement:** Not applicable.

**Data Availability Statement:** The last time step from the low-resolution ensemble stream function and example analysis code in Python is available on Github (doi:10.5281/zenodo.4667532). For more output files, please contact the corresponding author.

**Acknowledgments:** We thank William K. Dewar for his always insightful and fruitful discussion. The authors would like to acknowledge the developers of the Basilisk language (http://basilisk.fr/, accessed on 9 April 2021) upon which MSOM is based, and the developers of the xarray Python package [60], which we used to post process the model outputs.

**Conflicts of Interest:** The authors declare no conflict of interest. The funders had no role in the design of the study; in the collection, analyses or interpretation of data; in the writing of the manuscript, or in the decision to publish the results.

## Appendix A. Derivation of the Layered Quasi-Geostrophic Potential Vorticity

As the relative vorticity Equation (5) and layer-thickness Equation (6) have a common term on the right-hand side, they can be combined as:

$$\frac{D_i}{Dt}\zeta_{g;i} + \beta v_{g;i} = \frac{f_0}{H_i}\frac{D_i}{Dt}h_i, \tag{A1}$$

and we get the governing equation for QGPV $q_i = \zeta_{g;i} + \beta y - \frac{f_0}{H_i}h_i$ [7]. It is perhaps interesting to note that the QGPV remains identical for a stationary and temporally varying background stratification (viz. $g_1' = \frac{U^2}{H_1^\dagger}Fr_1^{-2}(t)$) although we have shown that this is not the case for the energy budget. The stream function is related to the layer displacement via $\eta_i = \frac{f_0}{g_i'}(\psi_{i+1} - \psi_i)$. The layer thickness can, therefore, be written using the stream function as [7]:

$$\begin{aligned} h_i &= H_i + \eta_{i-1} - \eta_i \\ &= H_i + \frac{f_0}{g_{i-1}'}(\psi_i - \psi_{i-1}) - \frac{f_0}{g_i'}(\psi_{i+1} - \psi_i), \end{aligned} \tag{A2}$$

where $\frac{D_1}{Dt}\eta_0 = \frac{D_3}{Dt}\eta_3 = 0$ due to rigid-lid and flat-bottom boundary conditions.

Now, suppose at any given time, we have total buoyancy ($B$) defined on a layer interface (Figure A1). Based on Taylor expansion, the layer interface displacement can be expanded as [7]:

$$\begin{aligned} \eta &= \left.\frac{\partial z}{\partial B}\right|_{z=H}[B_0(t, z=H) - B(t, z=H+\eta)] \\ &= -\left.\frac{\partial z}{\partial B}\right|_{z=H}b, \end{aligned} \tag{A3}$$

where $b = B - B_0$ is the QG fluctuation about the background buoyancy ($B_0$). Hence, we get:

$$\frac{b}{N^2} = -\eta, \tag{A4}$$

and taking the material derivative gives the buoyancy equation in the continuously stratified framework:

$$\frac{D}{Dt}\frac{b}{N^2} = -w. \tag{A5}$$

Equation (A4) gives the physical intuition that the material derivative of $b/N^2$ leads to vortex stretching.

**Figure A1.** Schematic of a relation between buoyancy ($B$) and layer interface displacement ($\eta$). The background buoyancy is $B_0$ defined at $z = H$.

## Appendix B. The Omega Equation with a Temporally Varying Background Stratification

We derive the QG omega equation using the continuously stratified framework. Taking the vertical derivative of the order-Rossby number momentum equations with the viscous term:

$$\partial_t u_{g;i} + u_{g;i}\partial_x u_{g;i} + v_{g;i}\partial_y u_{g;i} - f_0 v_{a;i} - \beta y v_{g;i} = -\partial_x \phi_{a;i}, \tag{A6}$$

$$\partial_t v_{g;i} + u_{g;i}\partial_x v_{g;i} + v_{g;i}\partial_y v_{g;i} + f_0 u_{a;i} + \beta y u_{g;i} = -\partial_y \phi_{a;i}, \tag{A7}$$

and multiplying them by $f_0$ gives:

$$\frac{D}{Dt}(f_0\partial_z u_g) + \partial_y \boldsymbol{u}_g \cdot \boldsymbol{\nabla}_{\mathrm{h}}b - f_0^2\partial_z v_a - \beta y f_0\partial_z v_g = -f_0\nu_4\partial_z \boldsymbol{\nabla}_{\mathrm{h}}^4 u_g, \tag{A8}$$

$$\frac{D}{Dt}(f_0\partial_z v_g) - \partial_x \boldsymbol{u}_g \cdot \boldsymbol{\nabla}_{\mathrm{h}}b + f_0^2\partial_z u_a + \beta y f_0\partial_z u_g = -f_0\nu_4\partial_z \boldsymbol{\nabla}_{\mathrm{h}}^4 v_g, \tag{A9}$$

and the horizontal gradients of the buoyancy Equation (A5) with the diffusive term yields:

$$\frac{1}{N^2}\frac{D}{Dt}\partial_x b + \partial_x b\partial_t \frac{1}{N^2} + \frac{\partial_x \boldsymbol{u}_g}{N^2}\cdot\boldsymbol{\nabla}_{\mathrm{h}}b + \partial_x w_a = -\kappa_4\partial_x \boldsymbol{\nabla}_{\mathrm{h}}^4 b, \tag{A10}$$

$$\frac{1}{N^2}\frac{D}{Dt}\partial_y b + \partial_y b\partial_t \frac{1}{N^2} + \frac{\partial_y \boldsymbol{u}_g}{N^2}\cdot\boldsymbol{\nabla}_{\mathrm{h}}b + \partial_y w_a = -\kappa_4\partial_y \boldsymbol{\nabla}_{\mathrm{h}}^4 b. \tag{A11}$$

Summing Equation (A8) with (A11), and $-$(A9) with (A10), and using the thermal wind relation, we get:

$$2\partial_y \boldsymbol{u}_g \cdot \boldsymbol{\nabla}_{\mathrm{h}}b + N^2\partial_y w_a - \beta y\partial_x b - f_0^2\partial_z v_a + \partial_y b N^2\partial_t \frac{1}{N^2} = 0. \tag{A12}$$

$$2\partial_x \boldsymbol{u}_g \cdot \boldsymbol{\nabla}_{\mathrm{h}}b + N^2\partial_x w_a + \beta y\partial_y b - f_0^2\partial_z u_a + \partial_x b N^2\partial_t \frac{1}{N^2} = 0. \tag{A13}$$

The viscous and diffusive terms do not appear as they cancel out due to the thermal–wind relation ($f_0\partial_z\zeta_g = \boldsymbol{\nabla}_{\mathrm{h}}^2 b$) and their parameters being set identical (viz. $\nu_4 = \kappa_4$ ($= Re_4^{-1}L^3 U$)). Taking $\partial_y$(A12) $+ \partial_x$(A13) gives the omega equation for a temporally varying background stratification:

$$N^2\boldsymbol{\nabla}_{\mathrm{h}}^2 w_a + f_0^2\partial_{zz}w_a = \beta\partial_x b - 2\boldsymbol{\nabla}_{\mathrm{h}}\cdot\mathbf{Q} - \boldsymbol{\nabla}_{\mathrm{h}}^2 b N^2\partial_t\frac{1}{N^2}. \tag{A14}$$

Although the last term on the right-hand side involves a time derivative, there is no time dependency in our case as we know the analytical form of the background stratification (Equation (4)). Its contribution to the omega equation turned out to be negligible (not shown).

## Appendix C. Decomposing the Mean and Eddy Energetics

In this section, we derive the mean and eddy KE equations. Equation (5) can be split into its mean and eddy component:

$$\frac{D^{\#}}{Dt}\boldsymbol{\nabla}_{\mathrm{h}}^2\overline{\psi} + \frac{D^{\#}}{Dt}\boldsymbol{\nabla}_{\mathrm{h}}^2\psi' + \boldsymbol{u}'_g\cdot\boldsymbol{\nabla}_{\mathrm{h}}[\boldsymbol{\nabla}_{\mathrm{h}}^2(\overline{\psi}+\psi')] + \beta\partial_x(\overline{\psi}+\psi') = f_0\partial_z(\overline{w}+w'), \tag{A15}$$

where $\frac{D^{\#}}{Dt} = \partial_t + \overline{\boldsymbol{u}_g} \cdot \boldsymbol{\nabla}_{\mathrm{h}}$. Multiplying this by $-\overline{\psi}$ gives:

$$\frac{D^{\#}}{Dt}\frac{|\boldsymbol{\nabla}_{\mathrm{h}}\overline{\psi}|^2}{2} - \boldsymbol{\nabla}_{\mathrm{h}} \cdot \overline{\boldsymbol{u}_g}\overline{\psi}\boldsymbol{\nabla}_{\mathrm{h}}^2\overline{\psi} - \overline{\psi}\frac{D^{\#}}{Dt}\boldsymbol{\nabla}_{\mathrm{h}}^2\psi' - \overline{\psi}\boldsymbol{u}_g' \cdot \boldsymbol{\nabla}_{\mathrm{h}}[\boldsymbol{\nabla}_{\mathrm{h}}^2(\overline{\psi} + \psi')]$$

$$- \beta\partial_x\frac{\overline{\psi}^2}{2} - \overline{\psi}\beta\partial_x\psi' = \overline{w}\overline{b} - \overline{\psi}f_0\partial_z w', \quad \text{(A16)}$$

and taking its ensemble mean yields the mean KE equation:

$$\frac{D^{\#}}{Dt}\frac{|\boldsymbol{\nabla}_{\mathrm{h}}\overline{\psi}|^2}{2} - \boldsymbol{\nabla}_{\mathrm{h}} \cdot \overline{\boldsymbol{u}_g}\overline{\psi}\boldsymbol{\nabla}_{\mathrm{h}}^2\overline{\psi} - \beta\partial_x\frac{\overline{\psi}^2}{2} = \overline{w}\overline{b} + \overline{\psi}\boldsymbol{\nabla}_{\mathrm{h}} \cdot \overline{\boldsymbol{u}_g'\boldsymbol{\nabla}_{\mathrm{h}}^2\psi'}. \quad \text{(A17)}$$

On the other hand, the ensemble mean of the total KE Equation (11) is:

$$\overline{\frac{D}{Dt}\frac{|\boldsymbol{\nabla}_{\mathrm{h}}\psi|^2}{2}} - \boldsymbol{\nabla}_{\mathrm{h}} \cdot \overline{\boldsymbol{u}_g\psi\boldsymbol{\nabla}_{\mathrm{h}}^2\psi} - \beta\partial_x\frac{\overline{\psi^2}}{2} = \overline{wb}, \quad \text{(A18)}$$

which can be expanded as:

$$\frac{D^{\#}}{Dt}\frac{|\boldsymbol{\nabla}_{\mathrm{h}}\overline{\psi}|^2}{2} + \frac{D^{\#}}{Dt}\frac{\overline{|\boldsymbol{\nabla}_{\mathrm{h}}\psi'|^2}}{2} + \boldsymbol{\nabla}_{\mathrm{h}} \cdot \overline{\boldsymbol{u}_g'\frac{|\boldsymbol{\nabla}_{\mathrm{h}}\psi'|^2}{2}} + \overline{\boldsymbol{u}_g' \cdot \boldsymbol{\nabla}_{\mathrm{h}}(\partial_x\overline{\psi}\partial_x\psi' + \partial_y\overline{\psi}\partial_y\psi')}$$

$$- \boldsymbol{\nabla}_{\mathrm{h}} \cdot \overline{\boldsymbol{u}_g\psi\boldsymbol{\nabla}_{\mathrm{h}}^2\psi} - \beta\partial_x\frac{\overline{\psi^2}}{2} = \overline{wb}. \quad \text{(A19)}$$

Taking the difference between Equations (A17) and (A19) gives the eddy KE equation:

$$\frac{D^{\#}}{Dt}\frac{\overline{|\boldsymbol{\nabla}_{\mathrm{h}}\psi'|^2}}{2} + \boldsymbol{\nabla}_{\mathrm{h}} \cdot \overline{\boldsymbol{u}_g'\frac{|\boldsymbol{\nabla}_{\mathrm{h}}\psi'|^2}{2}} + \boldsymbol{\nabla}_{\mathrm{h}} \cdot \overline{\boldsymbol{u}_g'(\partial_x\overline{\psi}\partial_x\psi' + \partial_y\overline{\psi}\partial_y\psi')}$$

$$- \boldsymbol{\nabla}_{\mathrm{h}} \cdot (\overline{\boldsymbol{u}_g\psi\boldsymbol{\nabla}_{\mathrm{h}}^2\psi} - \overline{\boldsymbol{u}_g}\overline{\psi}\boldsymbol{\nabla}_{\mathrm{h}}^2\overline{\psi}) - \beta\partial_x\frac{\overline{\psi'^2}}{2} = \overline{w'b'} - \overline{\psi}\boldsymbol{\nabla}_{\mathrm{h}} \cdot \overline{\boldsymbol{u}_g'\boldsymbol{\nabla}_{\mathrm{h}}^2\psi'}. \quad \text{(A20)}$$

Since the divergence terms vanish upon area integration, we can see the mean and eddy KE exchanging the term $-\overline{\psi}\boldsymbol{\nabla}_{\mathrm{h}} \cdot \overline{\boldsymbol{u}_g'\boldsymbol{\nabla}_{\mathrm{h}}^2\psi'}$ (Equations (A17) and (A20)). The same procedure can be done for Equation (6) or the buoyancy equation to derive the mean and eddy APE equations.

## Appendix D. The Three-Layer QG Lorenz Energy Cycle

The Lorenz energy cycle [2] for the first layer, dropping the divergence terms in Equations (A17) and (A20), while bringing back the viscous and diffusive terms becomes:

$$\partial_t K_1^{\#} = f_0\left[\overline{w_{a;1}\psi_1^{\dagger}} + H_1\overline{w_{a;1}}\frac{\overline{\psi_1} - \overline{\psi_2}}{H_2 + H_1}\right] + H_1\overline{\psi_1}\boldsymbol{\nabla}_{\mathrm{h}} \cdot \overline{\boldsymbol{u}_{g;1}'\boldsymbol{\nabla}_{\mathrm{h}}^2\psi_1'}$$

$$- \overline{\psi_1}\boldsymbol{\nabla}_{\mathrm{h}} \times \boldsymbol{\tau} + H_1\overline{\psi_1}\nu_4\boldsymbol{\nabla}_{\mathrm{h}}^4(\boldsymbol{\nabla}_{\mathrm{h}}^2\overline{\psi_1}), \quad \text{(A21)}$$

$$\partial_t \mathcal{K}_1 = f_0\left[\overline{w_{a;1}'\psi_1^{\dagger'}} + H_1\overline{w_{a;1}'\frac{\psi_1' - \psi_2'}{H_2 + H_1}}\right] - H_1\overline{\psi_1}\boldsymbol{\nabla}_{\mathrm{h}} \cdot \overline{\boldsymbol{u}_{g;1}'\boldsymbol{\nabla}_{\mathrm{h}}^2\psi_1'} + H_1\overline{\psi_1'\nu_4\boldsymbol{\nabla}_{\mathrm{h}}^4(\boldsymbol{\nabla}_{\mathrm{h}}^2\psi_1')}, \quad \text{(A22)}$$

$$\partial_t P_1^{\#} = -f_0\overline{w_{a;1}}(\overline{\psi_1} - \overline{\psi_2}) - \frac{f_0^2}{g_1'}(\overline{\psi_1} - \overline{\psi_2})\boldsymbol{\nabla}_{\mathrm{h}} \cdot \overline{\boldsymbol{u}_{g;1'}^{\dagger}(\psi_1' - \psi_2')}$$

$$- \frac{f_0^2}{g_1'}(\overline{\psi_1} - \overline{\psi_2})\kappa_4\boldsymbol{\nabla}_{\mathrm{h}}^4(\overline{\psi_1} - \overline{\psi_2}) - \frac{f_0^2(\overline{\psi_1} - \overline{\psi_2})^2}{2}\partial_t{g_1'}^{-1}, \quad \text{(A23)}$$

$$\partial_t \mathcal{P}_1 = -f_0 \overline{w'_{a;1}(\psi'_1 - \psi'_2)} + \frac{f_0^2}{g'_1}(\overline{\psi_1} - \overline{\psi_2})\boldsymbol{\nabla}_{\mathrm{h}} \cdot \overline{\boldsymbol{u}^\dagger_{g;1'}(\psi'_1 - \psi'_2)}$$

$$- \frac{f_0^2}{g'_1}\overline{(\psi'_1 - \psi'_2)\kappa_4\boldsymbol{\nabla}_{\mathrm{h}}^4(\psi'_1 - \psi'_2)} - \frac{f_0^2\overline{(\psi'_1 - \psi'_2)^2}}{2}\partial_t {g'_1}^{-1}. \quad \text{(A24)}$$

For the second layer:

$$\partial_t K_2^{\#} = f_0 \Big[ -(\overline{w_{a;1}}\,\overline{\psi_1^\dagger} - \overline{w_{a;2}}\,\overline{\psi_2^\dagger}) + H_2\big(\overline{w_{a;1}}\frac{\overline{\psi_1} - \overline{\psi_2}}{H_2 + H_1} + \overline{w_{a;2}}\frac{\overline{\psi_2} - \overline{\psi_3}}{H_3 + H_2}\big) \Big]$$

$$+ H_2\overline{\psi_2}\boldsymbol{\nabla}_{\mathrm{h}} \cdot \overline{\boldsymbol{u}'_{g;2}\boldsymbol{\nabla}_{\mathrm{h}}^2\psi'_2} + H_2\overline{\psi_2}\nu_4\boldsymbol{\nabla}_{\mathrm{h}}^4(\boldsymbol{\nabla}_{\mathrm{h}}^2\overline{\psi_2}), \quad \text{(A25)}$$

$$\partial_t \mathcal{K}_2 = f_0 \Big[ -(\overline{w'_{a;1}\psi_1^{\dagger\prime}} - \overline{w'_{a;2}\psi_2^{\dagger\prime}}) + H_2\big(\overline{w'_{a;1}\frac{\psi'_1 - \psi'_2}{H_2 + H_1}} + \overline{w'_{a;2}\frac{\psi'_2 - \psi'_3}{H_3 + H_2}}\big) \Big]$$

$$- H_2\overline{\psi_2}\boldsymbol{\nabla}_{\mathrm{h}} \cdot \overline{\boldsymbol{u}'_{g;2}\boldsymbol{\nabla}_{\mathrm{h}}^2\psi'_2} + H_2\overline{\psi'_2\nu_4\boldsymbol{\nabla}_{\mathrm{h}}^4(\boldsymbol{\nabla}_{\mathrm{h}}^2\psi'_2)}, \quad \text{(A26)}$$

$$\partial_t P_2^{\#} = -f_0\overline{w_{a;2}}(\overline{\psi_2} - \overline{\psi_3}) - \frac{f_0^2}{g'_2}(\overline{\psi_2} - \overline{\psi_3})\boldsymbol{\nabla}_{\mathrm{h}} \cdot \overline{\boldsymbol{u}^\dagger_{g;2'}}(\psi'_2 - \psi'_3) - \frac{f_0^2}{g'_2}(\overline{\psi_2} - \overline{\psi_3})\kappa_4\boldsymbol{\nabla}_{\mathrm{h}}^4(\overline{\psi_2} - \overline{\psi_3}), \quad \text{(A27)}$$

$$\partial_t \mathcal{P}_2 = -f_0\overline{w'_{a;2}(\psi'_2 - \psi'_3)} + \frac{f_0^2}{g'_2}(\overline{\psi_2} - \overline{\psi_3})\boldsymbol{\nabla}_{\mathrm{h}} \cdot \overline{\boldsymbol{u}^\dagger_{g;2'}(\psi'_2 - \psi'_3)} - \frac{f_0^2}{g'_2}\overline{(\psi'_2 - \psi'_3)\kappa_4\boldsymbol{\nabla}_{\mathrm{h}}^4(\psi'_2 - \psi'_3)}. \quad \text{(A28)}$$

For the third layer:

$$\partial_t K_3^{\#} = f_0 \Big[ -\overline{w_{a;2}}\,\overline{\psi_2^\dagger} + H_3\overline{w_{a;2}}\frac{\overline{\psi_2} - \overline{\psi_3}}{H_3 + H_2} \Big] + H_3\overline{\psi_3}\boldsymbol{\nabla}_{\mathrm{h}} \cdot \overline{\boldsymbol{u}'_{g;3}\boldsymbol{\nabla}_{\mathrm{h}}^2\psi'_3}$$

$$+ H_3\overline{\psi_3}[\nu_4\boldsymbol{\nabla}_{\mathrm{h}}^4(\boldsymbol{\nabla}_{\mathrm{h}}^2\overline{\psi_3}) + \epsilon\boldsymbol{\nabla}_{\mathrm{h}}^2\overline{\psi_3}], \quad \text{(A29)}$$

$$\partial_t \mathcal{K}_3 = f_0 \Big[ -\overline{w'_{a;2}\psi_2^{\dagger\prime}} + H_3\overline{w'_{a;2}\frac{\psi'_2 - \psi'_3}{H_3 + H_2}} \Big] - H_3\overline{\psi_3}\nabla_{\mathrm{h}} \cdot \overline{\boldsymbol{u}'_{g;3}\boldsymbol{\nabla}_{\mathrm{h}}^2\psi'_3}$$

$$+ H_3\overline{\psi'_3[\nu_4\boldsymbol{\nabla}_{\mathrm{h}}^4(\boldsymbol{\nabla}_{\mathrm{h}}^2\psi'_3) + \epsilon\boldsymbol{\nabla}_{\mathrm{h}}^2\psi'_3]}. \quad \text{(A30)}$$

Although the biharmonic diffusive terms in the APE Equations (A23), (A24), (A27) and (A28), which originate from diffusive terms in the layer-thickness Equation (6), are applied solely for numerical stability and their similarity with buoyancy in primitive equations, their formulation is conceptually similar to the Gent-McWilliams' skew diffusivity (GM; [61]). GM represents the process of baroclinic instability upon which isopycnal displacements are smoothed out adiabatically within the isopycnal layer. Considering the quasi two-dimensional and adiabatic nature of the QG system, the interpretation of layer-thickness diffusivity becomes similar to the GM skew diffusivity. A major difference here is that the diffusivity is set as the bihamonic diffusivity and as such, should be negligible in damping the resolved eddies [14,45].

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
