# Peer review of "The Seasonal Variability of the Ocean Energy Cycle from a Quasi-Geostrophic Double Gyre Ensemble"

_fluids, doi:10.3390/fluids6060206_

Round 1
Reviewer 1 Report
Review of Uchida et al. (2021)
Summary
This study aims at explaining the observed seasonal cycle of oceanic mesoscale kinetic energy. This seasonal cycle is often found to peak in summer, which has been explained in previous studies by an inverse energy cascade from the submesoscale (peaking in late winter/early spring) to the mesoscale. However, the authors show that a seasonal cycle of mesoscale kinetic energy with a maximum in summer can be simulated in a three-layer Quasi-Geostrophic double-gyre model configuration, whose forcing includes a seasonally varying stratification and which does not include submesoscale processes. To separate the mesoscale eddy component from the mean, the authors set up a 101-member ensemble of their model and define the ensemble mean to be the mean and the deviations from the ensemble mean to be the eddy contributions to the energy reservoirs. In fact, by diagnosing the Lorenz Energy Cycle (LEC) of the ensemble, the authors show that all four energy reservoirs of the LEC exhibit a maximum in summer. The summer maximum of mesoscale kinetic energy specifically, is explained by a stronger jet which is caused by stronger stratification in summer and generates stronger eddies compared to winter, when the stratification, the jet and the eddies shed from the jet, are weaker.
General Comments
- The major strength of this paper is the focus on the mesoscale. With submesoscale-permitting simulations becoming more and more feasible to conduct and more widely used, the fact that there are processes on the mesoscale that are not well or not at all understood is (in my point of view) often ignored by the modelling community. The authors show that the submesoscale is not necessarily required to explain the observed seasonal cycle of the mesoscale, which has been simulated before in ocean models but not been explained satisfactorily. This result is of profound importance, given that in the foreseeable future most global ocean models and especially coupled climate models will not resolve the submesoscale.
- The introduction gives a good overview of the topic, explaining the essential previous work that this study is based upon. However, it is missing any mention of previous studies (idealized or realistic model configuration) that investigated the seasonal cycle of mesoscale energy in simulations that do not permit submesoscale processes. Additionally, references to observations of the seasonal cycle of kinetic energy should be added along with the fact that the seasonal cycle of mesoscale kinetic energy does not peak in summer everywhere.
- In sections 2 and 3 (Model Description and Derivation of the Lorenz energy cycle) some variables are only defined in Table 1, while others are only defined in the text and some are not defined at all. Preferably (although this arguably a personal preference), all variables should be defined in the text for better and more fluid readability of the equations. Additionally, the derivation of the different equations is sometimes not completely clear, please see the specific comments for more details on this.
- The results are well presented but especially Figure 4 displays some variables that are not referenced in the text. While I understand that the authors want to display all terms of the derived budget, the figures would clearly benefit from some simplification. (One example: Fig 4c contains 6 terms, of which only one (Db) is referred to in the text. Additionally, all other terms do not seem to exhibit significant variability. Here, the authors could sum up the 5 other terms and have a plot with only 2 lines (Db and the sum of the others). I suggest that the authors carefully go through the figures again and try to simplify them or make then easier to read.
All figures would also benefit of a title that explains the general content of the figure at a first glance. - One aspect that should be added to the discussion is the possibility to transfer the presented results to other cases. E.g. What do the results of this study imply for regions away from the boundary currents/jets where barotropic instability probably plays a minor role etc.
Specific Comments
- the authors often write "while as" when only "while" is required
- l. 15: "in the" should be "of"
- l. 25-27: I propose to change these sentences to: "[...] energy cascade:
During summer, when the Western Boundary Current is stabilized and strengthened due to increased stratification, stronger mesoscale eddies are shed from the separated jet."
- l. 36: Please double-check reference 4.
- l. 49-50: I propose the following change: "[...] effect of gravity,
due to weak vertical stratification, has a similar magnitude as the effects of Earth's rotation."
- l. 58: it should be made clear that the first baroclinic Rossby radius of deformation is meant.
- l. 74: I propose to replace "time lag" by "time required"
- l. 77: I propose to replace ", namely" by ":"
- l. 81: An explanation what exactly in the definition of a QG model leads to the fact that it only resolves small Rossby number dynamics would be beneficial.
- l. 83: "QC" is defined to be "quasi-geostropic", but here "Quasi-geostrophy" is meant.
- l. 101-103: After stating (l. 81) that the QG model does not allow submesoscales, here the authors note that it does allow submesoscales. Please clarify.
- eq. 2 and 3: not all variables are defined (τ, y)
- l. 116-117: I propose to write: "[...]
perturb the first-layer stream function at a single, random grid point with a perturbation on the order [...]"
- l. 126: For readers from the OGCM-community, a translation to days would be pleasant.
- l. 134: The notation of ∂t etc. is not defined.
- l. 134: Φ is not defined.
- l. 144: It is not clear to me where the layer thickness equation is derived from. Please include a reference or a note on where the equation originates from.
- eq. 10: va;i , 2nd term on lhs, should be vg;i
- l. 150-151: It is not explained what QGPV is, additionally the step to derive the governing equation is not clear.
- l. 156: It is not clear to me, how the definition of the layer displacement is derived. Please include a reference or a note on where the equation originates from.
- l. 164-166: The viscous and diffusive terms have not been mentioned or defined before so it is a bit confusing that it is now stated that they cancel out.
- l. 182: wa should be wa;i in the last term on rhs
- eq. 21: double-check rhs terms: (Ψ2-Ψ3) or (Ψ3-Ψ2)
- l. 232: the references [39-41] are placed in such a way that I thought they would explain the eddies as intrinsic variability arising at QG scales, however, the references explain the concept of "response to external forcing" vs. "intrinsic variability". I suggest placing the references after "intrinsic variability" and find other suitable references for intrinsic variability at QG scales (if there are any).
- eq. 36: check rhs: (Ψ2-Ψ3) or (Ψ3-Ψ2)
- l. 275: Please clarify that the ensemble spread starts to grow one year after the perturbation, not one year after the start of the simulation.
- l. 318-319: It is not clear to me how a stabilized jet shedding stronger eddies should be the cause for more energy storage in summer? Maybe rephrase this to make it clearer.
- l. 381: It seems reference 43 is not the correct one. There is no realistic ocean simulation ensemble in this reference.
- eq. A24 and A25: Double check rhs (see comment on eq. 36 above)
Reviewer 2 Report
review attached

Round 2
Reviewer 2 Report
I am glad to know that the resolution does not affect the findings thus making them much more credible. I recommend to accept the paper in its present form.